# A reservoir bubble point pressure prediction model using the Adaptive Neuro-Fuzzy Inference System (ANFIS) technique with trend analysis

**Fahd Saeed Alakbari**[1]*, **Mysara Eissa Mohyaldinn**[1]*, **Mohammed Abdalla Ayoub**[1], **Ali Samer Muhsan**[2], **Ibnelwaleed A. Hussein**[3,4]

1 Petroleum Engineering Department, Universiti Teknologi PETRONAS, Bandar Seri Iskandar, Perak, Malaysia, 2 Mechanical Engineering Department, Universiti Teknologi PETRONAS, Bandar Seri Iskandar, Perak, Malaysia, 3 Gas Processing Center, College of Engineering, Qatar University, Doha, Qatar, 4 Department of Chemical Engineering, College of Engineering, Qatar University, Doha, Qatar

* fahd_19001032@utp.edu.my (FSA); mysara.eissa@utp.edu.my (MEM)

## Abstract

The bubble point pressure ($P_b$) could be obtained from pressure-volume-temperature (PVT) measurements; nonetheless, these measurements have drawbacks such as time, cost, and difficulties associated with conducting experiments at high-pressure-high-temperature conditions. Therefore, numerous attempts have been made using several approaches (such as regressions and machine learning) to accurately develop models for predicting the $P_b$. However, some previous models did not study the trend analysis to prove the correct relationships between inputs and outputs to show the proper physical behavior. Thus, this study aims to build a robust and more accurate model to predict the $P_b$ using the adaptive neuro-fuzzy inference system (ANFIS) and trend analysis approaches for the first time. More than 700 global datasets have been used to develop and validate the model to robustly and accurately predict the $P_b$. The proposed ANFIS model is compared with 21 existing models using statistical error analysis such as correlation coefficient (R), standard deviation (SD), average absolute percentage relative error (AAPRE), average percentage relative error (APRE), and root mean square error (RMSE). The ANFIS model shows the proper relationships between independent and dependent parameters that indicate the correct physical behavior. The ANFIS model outperformed all 21 models with the highest R of 0.994 and the lowest AAPRE, APRE, SD, and RMSE of 6.38%, -0.99%, 0.074 psi, and 9.73 psi, respectively, as the first rank model. The second rank model has the R, AAPRE, APRE, SD, and RMSE of 0.9724, 9%, -1.58%, 0.095 psi, and 13.04 psi, respectively. It is concluded that the proposed ANFIS model is validated to follow the correct physical behavior with higher accuracy than all studied models.

**Data Availability Statement:** All relevant data are within the manuscript and its Supporting Information files.

**Funding:** The authors would like to give their heartfelt thanks to the Yayasan Universiti Teknologi PETRONAS (YUTP FRG Grant No: 015LC0-226) at Universiti Teknologi PETRONAS for supporting this study. The funder is the supervisor of the first author of the paper. He has major roles in supervision and funding, problem conceptualization, drafting, and reviewing.

**Competing interests:** The authors have declared that no competing interests exist.

**Abbreviations:** ANFIS, adaptive neuro-fuzzy inference system; ANN, artificial neural networks; FL, fuzzy logic; SAS, statistical analysis system; PVT, pressure-volume-temperature; AAE, average absolute error; APRE, average percent relative error; AAPRE, average absolute percent relative error; RMSE, root mean square error; SD, standard deviation; USA, United States of America; MS-Excel, Microsoft Excel; TrA, trend analysis; MMP, minimum miscibility pressure;

**Latin synonyms**

Pb, bubble point pressure, psi; R, correlation coefficient; R2, coefficient of determination; $E_{max.}$, maximum absolute percent relative error; $E_{min.}$, minimum absolute percent relative error; API, oil API gravity,°API; Rs, gas solubility, SCF/STB; $T_f$, reservoir temperature,°F; Bob, oil formation volume factor at the bubble point pressure, bbl/STB; Y, the normalized parameter; $Y_{max}$, the maximum normalized value (1); $Y_{min}$, the minimum normalized value (-1); X, the input variable; $X_{min}$, the minimum of the variable; $X_{max}$, the maximum of the variable;

**Greek synonyms**

$\gamma_g$, gas-specific gravity; $\gamma_o$, oil-specific gravity.

# 1. Introduction

Determination or measurement of an accurate reservoir bubble point pressure ($P_b$) is essential for achieving accurate reservoir and petroleum production calculations [1–4]. As a result, obtaining the $P_b$ with high accuracy is necessary.

Numerous researchers studied the $P_b$ for different crude oils. In North America, Standing [5], Lasater [6], Glaso [7], Petrosky and Farshad [8], De Ghetto et al. [9], Velarde et al. [4], and Dindoruk and Christman [10] showed correlations applied to determine the $P_b$ based on Rs, $\gamma_g$, API, and $T_f$. Standing [5] and Lasater [6] utilized 105 and 158 datasets from the USA and Canada to develop their models. Glaso [7] applied some regressions methods to create a correlation for Pb with a standard deviation (SD) of 6.98. Petrosky and Farshad [8] used 90 Gulf Mexico datasets to develop their $P_b$ model by applying regression methods (involving Statistical Analysis System (SAS) software). De Ghetto et al. [9] and Velarde et al. [4] used regressions techniques to create their equations to determine the $P_b$, and they mentioned that their correlations have AAE of 12.8% and 11.7%. Dindoruk and Christman [10] showed a correlation employed to determine the $P_b$ using 100 datasets and MS-Excel software.

Al-Marhoun [11], Dokla and Osman [12], Almehaideb [13], Mehran et al. [14], Bolondarzadeh et al. [15], Hemmati and Kharrat [16], Mazandarani and Asghari [17], Khamehchi et al. [18], and Gomaa [19] developed their $P_b$ correlations depended on the Middle East crude oils. Al-Marhoun [11] utilized Rs, $\gamma_g$, API, and $T_f$ as independent parameters to create a correlation to determine the $P_b$ by applying the non-linear multiple regression method using 160 data points. Dokla and Osman [12] and Almehaideb [13] displayed $P_b$ correlations using 51 and 62 data points from the United Arab Emirates, and their equations have AAE of 7.61% and 4.997%, respectively. Mehran et al. [14], Bolondarzadeh et al. [15], Hemmati and Kharrat [16], Mazandarani and Asghari [17], Khamehchi et al. [18] operated regression methods to create their $P_b$ equations using datasets from Iranian fields. Gomaa [19] developed the correlation based on Rs, $\gamma_g$, API, and Tf and disclosed that their equation has the AAE and the SD of 8.12% and 10.69.

In Africa, Macary and EL-Batanoney [20] showed an equation used to predict the $P_b$ with AAE of 7.04% using Rs, $\gamma_g$, API, and $T_f$ as independent variables and 90 datasets from Egypt. Hanafy et al. [21] used only the Rs as input parameter, the regression methods, and 324 datasets from Egyptian fields to determine the $P_b$. Sharrad and Abd-Alrahman [22] found a $P_b$ equation using more than thirty Libyan datasets and EViews software and displayed their correlation with the AAE of 8.7%.

Frashad et al. [23] showed the $P_b$ correlation with SD of 37.02 using regression methods and 43 datasets from Colombia. Omar and Todd [24] applied non-linear regression analysis and more than ninety Malaysian datasets to display their $P_b$ correlation and indicated that the correlation has AAE and SD of 7.17% and 9.54.

Vasquez and Beggs [25], Kartoatmodjo and Schmidt [26], Al-Shammasi [27], and Arabloo et al. [28] proposed equations for predicting the $P_b$ based on Rs, $\gamma_g$, API, and Tf and utilizing data points from different places. Kartoatmodjo and Schmidt [26] employed more than 5000 datasets from different regions in North America and used a regression approach to build the $P_b$ correlation with 20.17% (AAE). Al-Shammasi [27] utilized a regression approach, 1661 datasets from different places to develop a $P_b$ correlation, and stated that the correlation could predict the $P_b$ with 17.849% AAE and 17.16 SD. Arabloo et al. [28] represented a $P_b$ correlation with an AAE of 18.9, operating LINGO software and more than 700 global datasets. Fig 1 illustrates the previously published models based on used data locations.

Nowadays, machine learning and deep learning methods are used to develop the $P_b$ model. Alakbari et al. [30] used artificial neural networks and fuzzy logic approaches for predicting

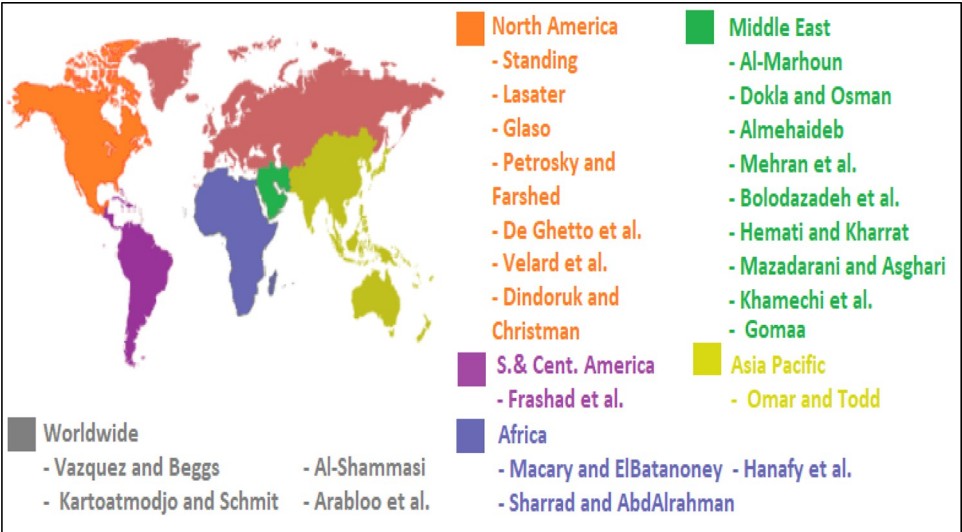

**Fig 1. Previous models based on used data locations [recreated from copyright free open source [29]].**

the $P_b$ based on Rs, $\gamma_g$, API, and Tf. Yang et al. [31] represented a correlation that can be used to predict the $P_b$ using some artificial intelligent algorithms, namely neural networks. Alakbari et al. [32] created their model based on the Rs, $\gamma_g$, API, and Tf as inputs and more than 700 datasets, and they showed that their model has the absolute average percent relative error and the (R) were 8.422% and 0.990. Nonetheless, the previous models are required to improve their accuracy in obtaining the $P_b$.

Numerous researchers successfully applied the adaptive neuro-fuzzy inference system (ANFIS) method in engineering calculations. A noise assessment of wind turbine was predicted using the ANFIS [33]. The ionic and electronic conductivity of materials was estimated utilizing the ANFIS [34]. Ayoub et al. [35] developed a model to obtain the drilling rate of penetration using the ANFIS technique. The wind power density was determined by applying the ANFIS [36]. Sambo et al. [37] used ANFIS to determine water saturation from seismic attributes. Hamdi and Chenxi [38] proposed an ANFIS model to predict $CO_2$ minimum miscibility pressure (MMP) with higher accuracy. A recent study has applied ANFIS to model the isothermal oil compressibility below the $P_b$ Ayoub et al. [39].

This research aims to build a robust and higher accurate model that can be used to determine the $P_b$ using the ANFIS method with the trend analysis (TrA). The only attempt to apply ANFIS for developing $P_b$ correlations is the one proposed by Shojaei et al. [40], who used 750 data points to build the $P_b$ model. However, they have not studied the TrA to prove the proper physical behavior for their model. Therefore, in this study, a robust and highly accurate ANFIS model was developed to predict the $P_b$ through TrA. More than 700 global datasets and the ANFIS method were applied with the trend analysis that is used to find the relationships between the independent variables (Rs, $\gamma_g$, API, and $T_f$) and dependent variable ($P_b$) to indicate the correct physical behavior to build our ANFIS model with the trends analysis that is used for the first time to a robustly and accurately determine the $P_b$. Moreover, statistical error analyses such as R were utilized to compare the ANFIS and all existing models' accuracy.

## 2. Methodology

### 2.1 Data collection and pre-processing

More than seven hundred data sets were gathered from existing sources [11, 24, 28] to build the proposed ANFIS model. The Rs, $\gamma_g$, API, and Tf are utilized as independent parameters in this study because most of the studies in the literature consider these parameters as inputs; however, Hanafy et al. [21] used only the Rs as the input to predict the $P_b$, Table 1. Furthermore, the (R) for independent parameters (Rs, $\gamma_g$, API, and $T_f$) to the dependent parameter ($P_b$) was found to evaluate the importance of the independent and dependent parameters as shown in Fig 2. From this figure, we can see the (R) of 0.876 for the Rs, and the $P_b$ means that the $P_b$ can be a strong function of the Rs. As displayed in Fig 2, the (R) of -0.513 for the $\gamma_g$ and

**Table 1. Comparison of input parameters used in the published correlations and the proposed ANFIS model.**

| No | Model | Input parameters | | | | |
|----|-------|------------------|---|---|---|---|
| | | Bubble point oil volume factor (Bob) (bbl/STB) | Gas to oil ratio (Rs) (scf/STB) | Gas-specific gravity ($\gamma_g$) | Oil-specific gravity (API) (°API) | Reservoir temperature ($T_f$) (°F) |
| 1 | Standing (1947) [5] | | √ | √ | √ | √ |
| 2 | Lasater (1958) [6] | | √ | √ | √ | √ |
| 3 | Glaso (1980) [7] | | √ | √ | √ | √ |
| 4 | Vazquez and Beggs (1980) [25] | | √ | √ | √ | √ |
| 5 | Al-Marhoun (1988) [11] | | √ | √ | √ | √ |
| 6 | Kartoatmodjo and Schmit (1991) [26] | | √ | √ | √ | √ |
| 7 | Dokla and Osman (1992) [12] | | √ | √ | √ | √ |
| 8 | Petrosky and Farshed (1993) [8] | | √ | √ | √ | √ |
| 9 | Macary and El-Batanoney (1993) [20] | | √ | √ | √ | √ |
| 10 | Omar and Todd (1993) [24] | √ | √ | √ | √ | √ |
| 11 | De Ghetto et al. (1994) [9] | | √ | √ | √ | √ |
| 12 | Frashad et al. (1996) [23] | | √ | √ | √ | √ |
| 13 | Almehaideb (1997) [13] | √ | √ | √ | √ | √ |
| 14 | Hanafy et al. (1997) [21] | | √ | | | |
| 15 | Velarde et al. (1997) [4] | | √ | √ | √ | √ |
| 16 | Al-Shammasi (1999) [27] | | √ | √ | √ | √ |
| 17 | Dindoruk and Christman (2001) [10] | | √ | √ | √ | √ |
| 18 | Mehran et al. (2006) [14] | | √ | √ | √ | √ |
| 19 | Bolondarzadeh et al. (2006) [15] | | √ | √ | √ | √ |
| 20 | Hemati and Kharrat (2007) [16] | √ | √ | √ | √ | √ |
| 21 | Mazandarani and Asghari (2007) [17] | | √ | √ | √ | √ |
| 22 | Khamechchi et al. (2009) [18] | | √ | √ | √ | √ |
| 23 | Arabloo et al. (2014) [28] | | √ | √ | √ | √ |
| 24 | Gomaa (2016) [19] | | √ | √ | √ | √ |
| 25 | Sharrad and Abd-Alrahman (2019) [22] | | √ | √ | √ | √ |
| 26 | Proposed ANFIS | | √ | √ | √ | √ |

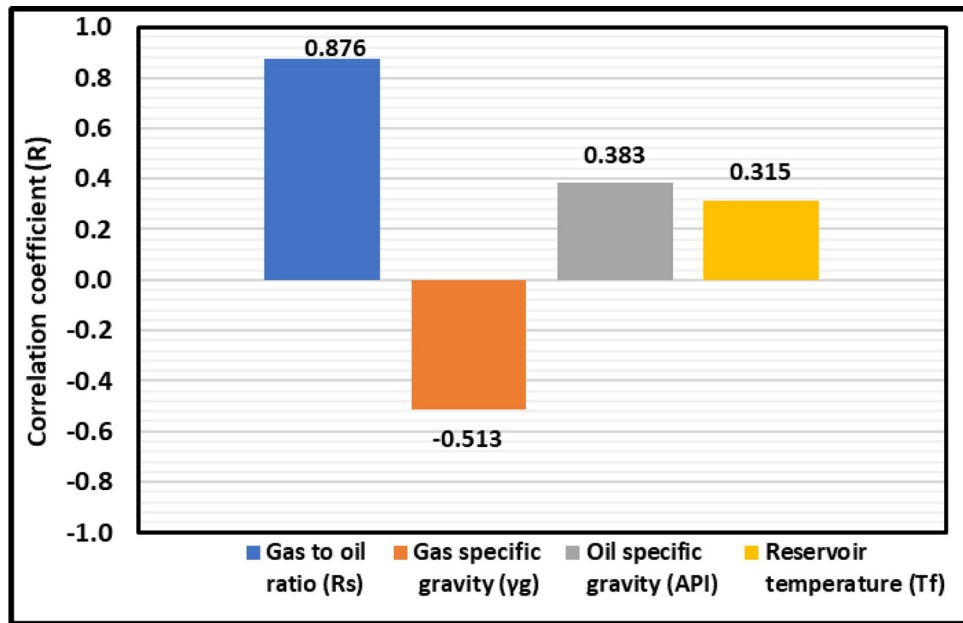

**Fig 2. Relative importance of inputs with P_b output.**

the $P_b$ indicates that the $P_b$ can be a moderate function of the $\gamma_g$ and the (R) of 0.383 and 0.315 for the API and $T_f$ proves that the $P_b$ can be a weak function of the API and $T_f$.

Before the ANFIS model was applied, the collected data were split into two parts 70% for training the model and 30% for testing the proposed ANFIS model. The statistical description of the training and testing datasets is shown in Table 2. As in the table, the training and testing datasets are at the same ranges to build and evaluate the ANFIS model with the same data ranges. It is essential to avoid the over-fitting and under-fitting issues; data randomization was used to overcome these issues. In addition, all parameters for the training and testing datasets were normalized between -1 and 1 to scale them in this range based on the following equation:

$$Y = (Y_{max} - Y_{min}) \times (X - X_{min})/(X_{max} - X_{min}) + Y_{min} \qquad (1)$$

Where:
Y: the normalized parameter.
$Y_{max}$: the maximum normalized value (1).
$Y_{min}$: the minimum normalized value (-1).
X: the input variable.
$X_{min}$: the minimum of the variable.
$X_{max}$: the maximum of the variable.

**Table 2. Statistical description of the data.**

| Parameters | Training data | | | Testing data | | |
|---|---|---|---|---|---|---|
| | Minimum | Maximum | SD | Minimum | Maximum | SD |
| Bubble point pressure (P_b) psi | 126 | 7127 | 1151.55 | 130 | 4432 | 1135.4 |
| Gas to oil ratio (Rs) SCF/STB | 9 | 2637 | 423.50 | 26 | 1850 | 424.93 |
| Gas-specific gravity ($\gamma_g$) | 0.5890 | 1.367 | 0.1593 | 0.5890 | 1.367 | 0.1622 |
| Oil-specific gravity (API) °API | 15.30 | 59.50 | 7.32 | 19.40 | 51.70 | 6.38 |
| Reservoir temperature ($T_f$) °F | 74 | 294 | 49.46 | 74 | 271 | 45.36 |

## 2.2 Proposed ANFIS model strategy

ANFIS is a combination of artificial neural networks (ANN) and fuzzy logic (FL), and it is one of the neural networks that use the Takagi-Sugeno fuzzy inference system. The Takagi-Sugeno fuzzy model applies two fuzzy rules [41]:

rule 1: if ($x_1$ is $A_1$) and ($x_2$ is $B_1$), then Eq (2) is used.

$$f_1 = p_1 x_1 + q_1 x_2 + r_1 \qquad (2)$$

rule 2: if ($x_1$ is $A_2$) and ($x_2$ is $B_2$), then Eq (3) is applied.

$$f_2 = p_2 x_1 + q_2 x_2 + r_2 \qquad (3)$$

where:

$x_1$ and $x_2$: inputs.

$A_1$, $A_2$, $B_1$, and $B_2$: membership values.

$p_1$, $q_1$, $r_1$, $p_2$, $q_2$, and $r_2$: parameters of the output functions $f_1$ and $f_2$, respectively.

As displayed in Fig 3, the ANFIS structure is constructed of five layers. These layers are the fuzzification layer, rule layer, normalization layer, defuzzification layer, and output layer. ANFIS is a multilayer feedforward neural network with supervised learning capability (a hybrid learning rule) [42, 43]. For the Sugeno fuzzy reasoning, the default defuzzification technique was applied. It can be a weighted average of all rule outputs. The fuzzified input values can be an algebraic sum of consequent fuzzy sets for the used aggregate technique. Firstly, input characteristics transfer to input membership functions. Then, they move to rules. After that, they shift to a set of output characteristics. Next, they go to output membership functions. Finally, the output membership functions provide output [44].

The ANFIS technique has advantages of showing better results than other methods. The ANFIS shows a better learning ability. It can perform a highly non-linear mapping. It has fewer adjustable parameters than those needed in other machine learning. Its structure can allow for parallel computation. Its networks show a well-structured knowledge representation and can also allow better integration with other control design methods [45]. ANFIS can combine ANN and Fl in a single tool to make the technique superb in reaching a quicker decision about the mapped relationship between the feature and target parameters [46]. The ANFIS has

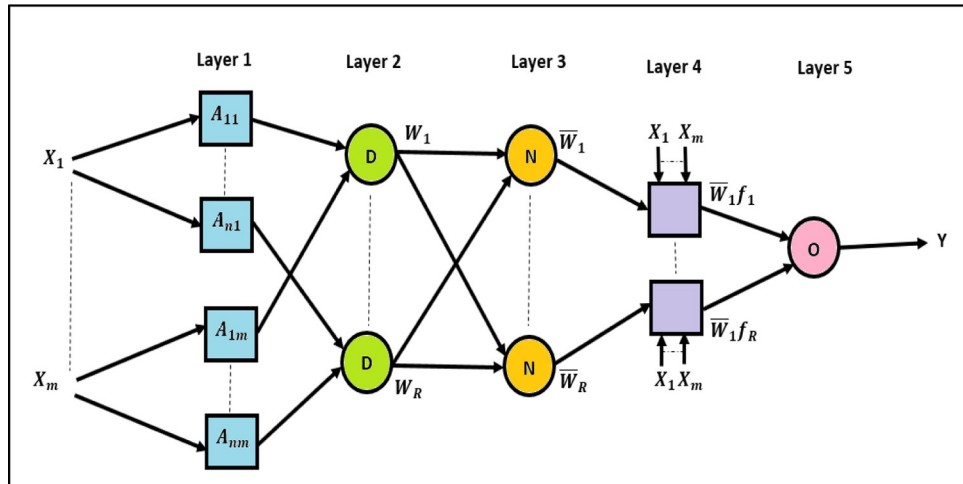

**Fig 3. The workflow of MATLAB ANFIS structure.**

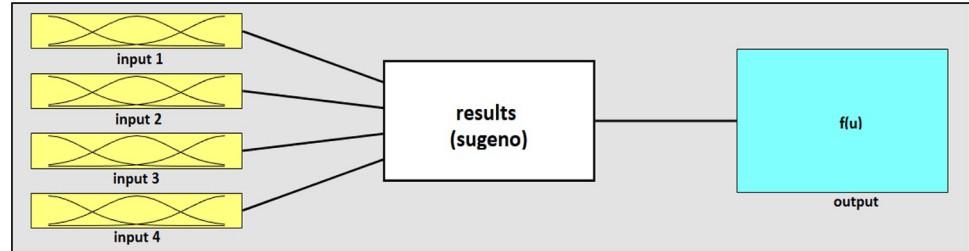

**Fig 4. ANFIS system results with four input parameters, three rules, and one output, (generated from MATLAB R2019b).**

the benefit of decreased training time not only because of its smaller dimensions but also because the network is initialized with parameters in relation to the problem domain [47].

The proposed ANFIS model in this work was built using MATLAB R2019b. Fig 4 demonstrates the ANFIS output generated from MATLAB 2019b. The type of membership function applied in this proposed ANFIS model is Gaussian curve membership. The optimal hyperparameters of ANFIS were selected by using the manual method. In the manual method, each parameter changed in its different types or values. Then, the model accuracy and the correct trend analysis were checked. Finally, the optimal hyperparameters were selected with the proper trend analysis for the highest accuracy, as shown in Table 3.

## 3. Results and discussion

The ANFIS model was evaluated by conducting two tests. The proposed ANFIS model was first investigated by conducting TrA to ensure that all inputs follow the proper physical behavior. After that, the ANFIS model and studied correlations were compared. Statistical error analysis, namely, (R), standard deviation (SD), average percent relative error (APRE), average absolute percentage relative error (AAPRE), and root mean square error (RMSE), were performed to show the performance of the ANFIS and studied models.

### 3.1 Trend analysis (TrA)

The trend analysis (TrA) can be used to study the reliability of models. TrA can be applied by changing the studied input between the minimum and maximum values while keeping the

**Table 3. Descriptions of the optimal ANFIS model hyperparameters.**

| Parameter | Description/value |
|---|---|
| Fuzzy structure | Sugeno-type |
| Initial FIS for training | genfis2 |
| Membership function type | Dsigmf |
| Output membership function | Linear |
| Cluster centre's range of influence | 0.459 |
| Number of inputs | 4 |
| Number of outputs | 1 |
| Optimization method | Hybrid |
| Number of fuzzy rules | 10 |
| Training epoch number | 24 |
| Initial step size | 0.3555 |
| Step size decrease rate | 0.2 |
| Step size increase rate | 2 |

other inputs at their constant mean values. The studied input, such as Rs, is plotted as the x-axis and the output $P_b$ as the y-axis [27, 48–50]. The TrA is an essential part of this work, as some researchers used ANFIS, but they have not applied the trend analysis [40]. Without considering the trend analysis, it was clear that the ANFIS model may show fake high accuracy. As a result, the models developed without considering the trend analysis should not be considered as a reliable tool.

The trend analysis was conducted for the ANFIS, and 21 studied models to study the relationships between the inputs (Rs, $\gamma_g$, API, $T_f$) and output $P_b$ to show the physical behavior.

In the TrA study, the four independent variables (Rs, $\gamma_g$, API, $T_f$) were selected because most previous models used these variables; nevertheless, the oil formation volume factor was not considered in our model because it is only utilized by [13, 16, 24]. The TrA was performed to represent the proper relationships between the Rs, $\gamma_g$, API, $T_f$ and the $P_b$ to show the actual physical behavior for the studied parameters and validated the ANFIS model.

Fig 5 presents the Rs TrA for the ANFIS and all existing models. As shown in Fig 5, the ANFIS and all the previous models show the proper relationships between the Rs and the $P_b$. Increasing the Rs increases the $P_b$. However, Farshad's [23] and Almehaideb's [13] correlations indicate that the $P_b$ was -812.6 and -207.5 psi at Rs 26 SCF/STB (as shown in Fig 5) because they built their correlation based on Rs ranges from 217 to 1406 and from 128 to 3871 SCF/STB, respectively. Fig 6 indicates that the developed ANFIS model follows the proper relationships between the Rs and the $P_b$ to correct physical behavior. Li et al. [51] showed that increasing the Rs increased the $P_b$.

The TrA of $\gamma_g$ for the ANFIS and all current models is demonstrated in Fig 7. The ANFIS and most existing models revealed that the $\gamma_g$ is inversely proportional to the $P_b$, which proves the proper relationships between the $\gamma_g$ and the $P_b$; nevertheless, Hanafy et al.'s [21] correlation displayed that changing the $\gamma_g$ does not change the $P_b$ as indicated by the constant trend. This indicates an incorrect relationship between the $\gamma_g$ and the $P_b$ because $\gamma_g$ was not considered as

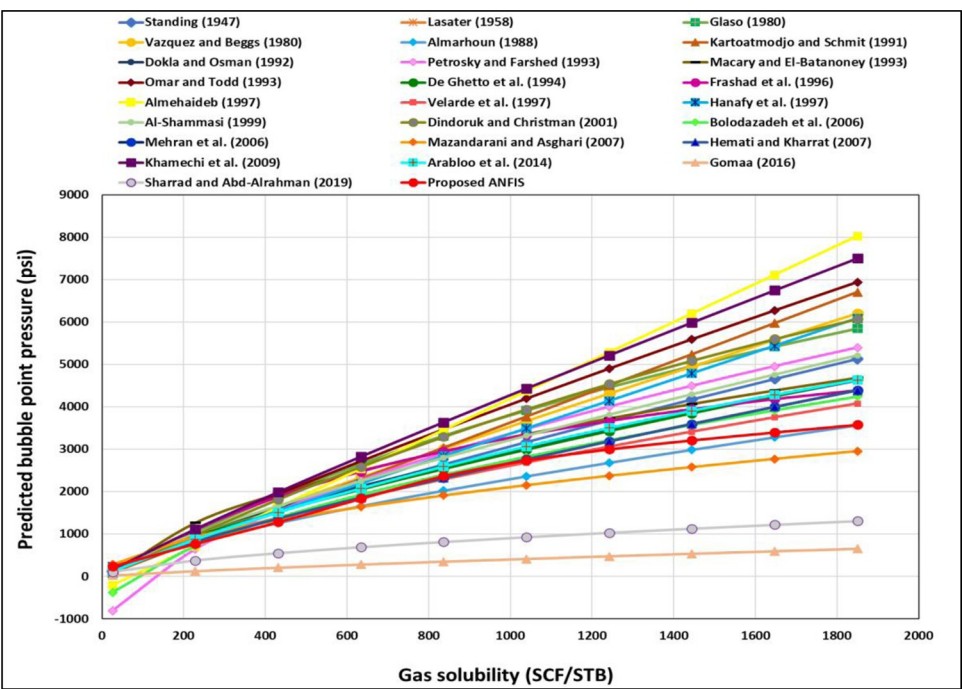

**Fig 5. Rs TrA of the ANFIS and existing models.**

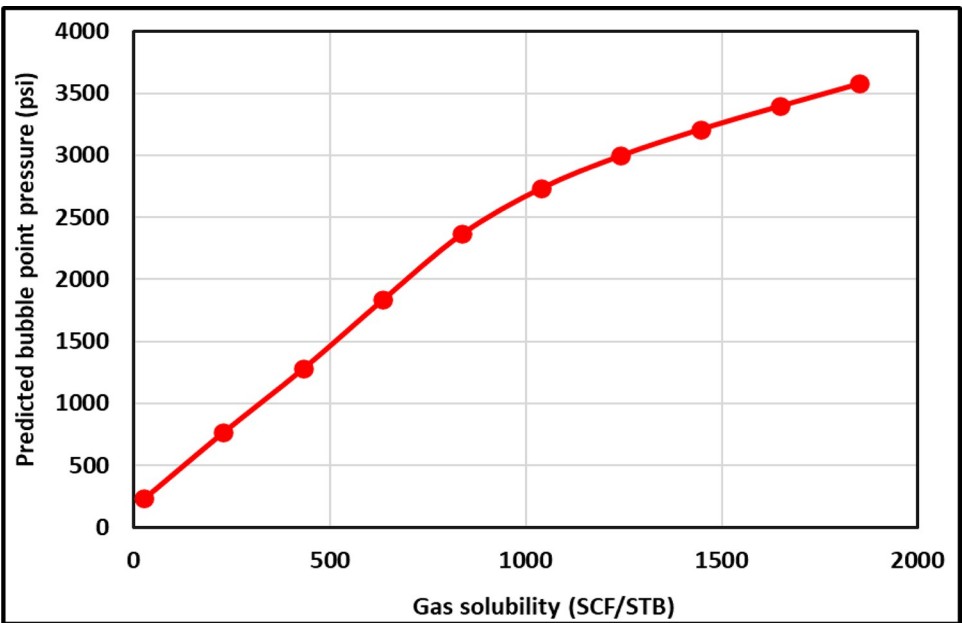

**Fig 6. Rs TrA of proposed ANFIS model.**

input in their model. Goma's [19] correlation showed that the $P_b$ was slightly increased by increasing the $\gamma_g$ and the correlation indicate improper TrA for $\gamma_g$. Omar and Todd's [24] correlation represented that the $P_b$ decreases and then increases by increasing the $\gamma_g$, which is also improper relationships between the $\gamma_g$ and the $P_b$. Therefore, Omar and Todd's [24], Hanafy et al.'s [21], and Goma's [19] models represent incorrect relationships between the $\gamma_g$ and the $P_b$, and hence, improper physical behavior for $\gamma_g$ trend. Fig 8 illustrated the correct trend $\gamma_g$ for the ANFIS model. Al-Shammasi [27] proved that growing the $\gamma_g$ declines the $P_b$.

Fig 9 shows the TrA of API for the ANFIS and all current models. The ANFIS and most models display the proper relationships between the API and the $P_b$. The higher the API, the lower the $P_b$ is (Fig 9); however, Dokla and Osman [12], Hanafy et al. [21], and Gomaa [19] models do not show the correct relationships between the API and the $P_b$, indicating incorrect physical behavior. Dokla and Osman's [12] correlation showed that the $P_b$ was slightly decreased by rising the API, (Fig 9). Gomaa's [19] correlation demonstrated that increasing the API also drops the $P_b$ slightly (Fig 9). Hanafy et al.'s [21] equation displayed that the $P_b$ is constant with changing the API (Fig 9). Petrosky and Farshad's [8] correlation shows that the $P_b$ is -37.37 psi and -145.91 psi at 48.11 and 51.7˚API, Fig 9 because they developed the equation in (16.3–45˚API) range. The ANFIS model presents the correct relationships between the API and the $P_b$, indicating proper physical behavior, as shown in Fig 10. Al-Shammasi [27] also revealed that increasing the API drops the $P_b$.

The TrA of the $T_f$ for the ANFIS and all current models is illustrated in Fig 11. As shown in Fig 11, the ANFIS and most current models follow the proper relationships between the $T_f$ and the $P_b$, increasing the $T_f$ increases the $P_b$; nonetheless, Dokla and Osman's [12] equation indicates that the $P_b$ declines by increasing $T_f$ indicating incorrect relationships between the $T_f$ and the $P_b$. Hanafy et al.'s [21] correlation also displays a constant $P_b$ with increasing the $T_f$ to indicate the improper relationships between the $T_f$ and the $P_b$. Dindoruk and Christman's [10] and Arabloo et al.'s [28] correlations represent that the $P_b$ is slightly changed by growing the $T_f$ to show incorrect physical behavior for the $T_f$ trend. The correct $T_f$ trend for the proposed

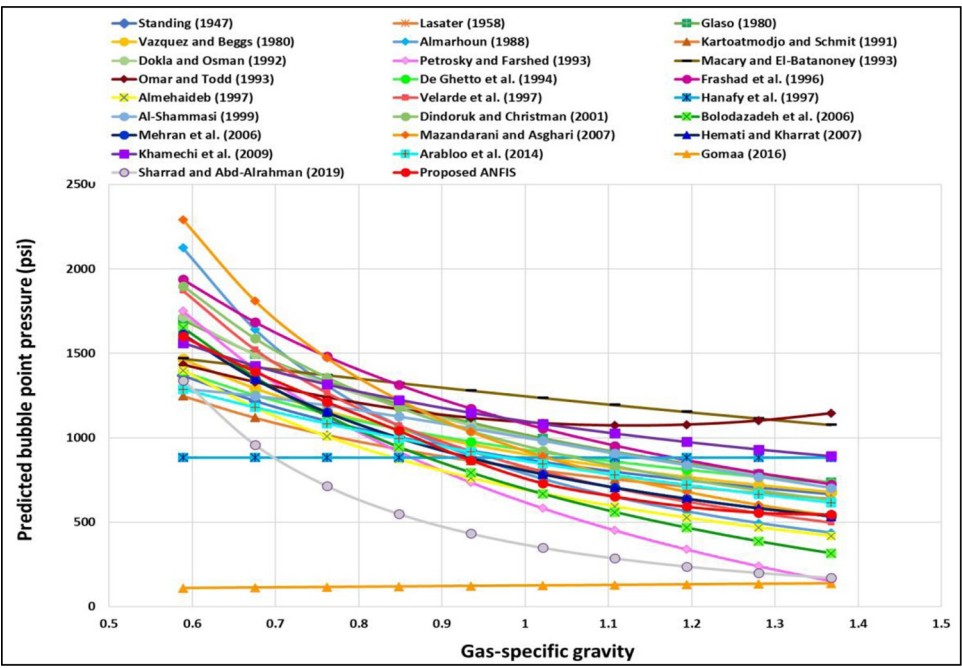

**Fig 7. $\gamma_g$ TrA of the ANFIS and existing models.**

ANFIS model is clearly represented in Fig 12. The temperature can drop the gas density; therefore, the temperature can increase the $P_b$.

From the TrA study, we can conclude that all independent parameters (Rs, $\gamma_g$, API, $T_f$) of the ANFIS model represent the proper relationships with the $P_b$ to indicate the correct physical behavior; however, Dokla and Osman's [12], Omar and Todd's [24], Hanafy et al.'s [21], and

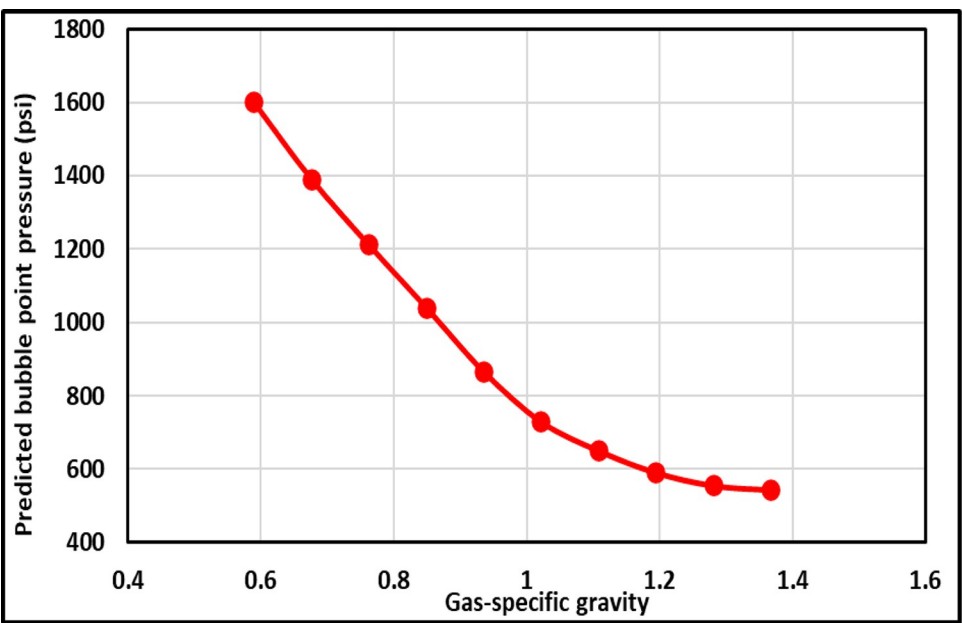

**Fig 8. $\gamma_g$ TrA of proposed ANFIS model.**

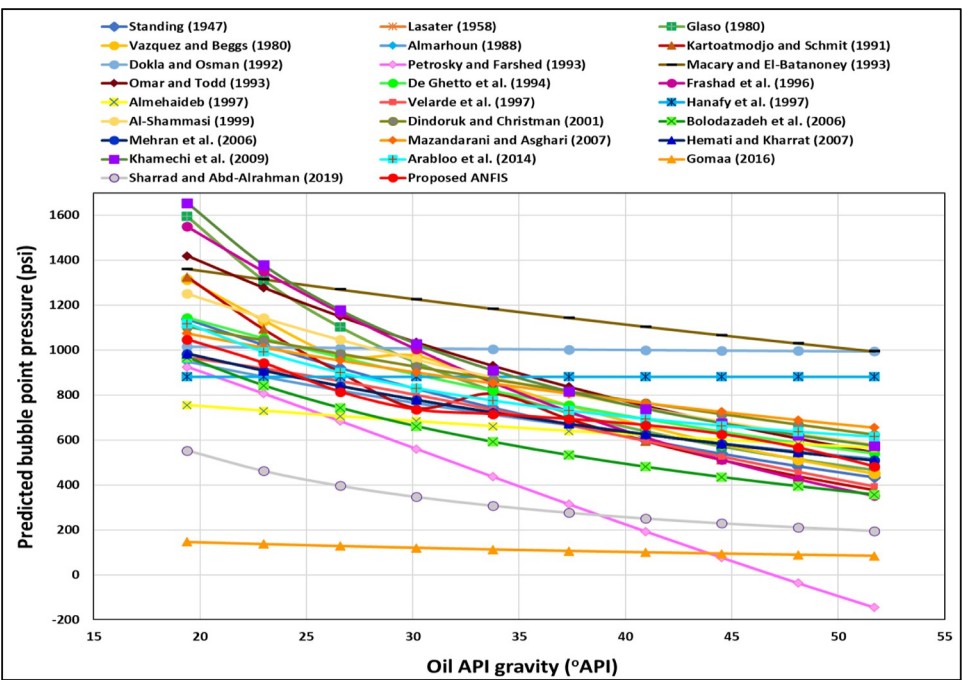

**Fig 9. API TrA of the ANFIS and existing models.**

Goma's [19] correlation show the improper relationships between the independent parameters and the $P_b$ to indicate the incorrect physical behavior. Petrosky and Farshad's [8] and Almehaideb's [13] correlations display some negative $P_b$ because the Rs and API as inputs for these negative values do not include in their study ranges.

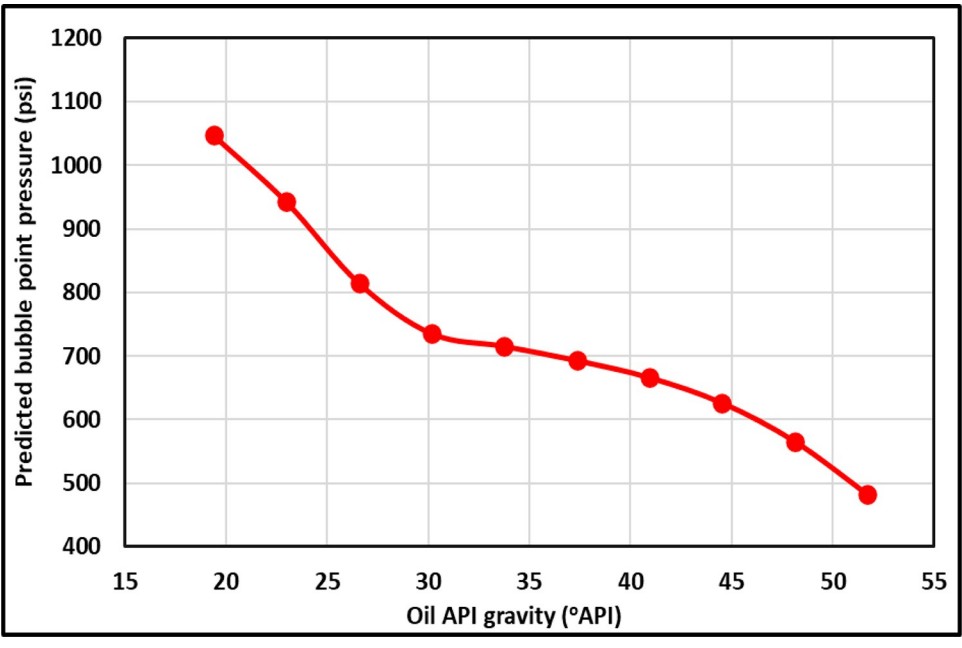

**Fig 10. API TrA of the ANFIS model.**

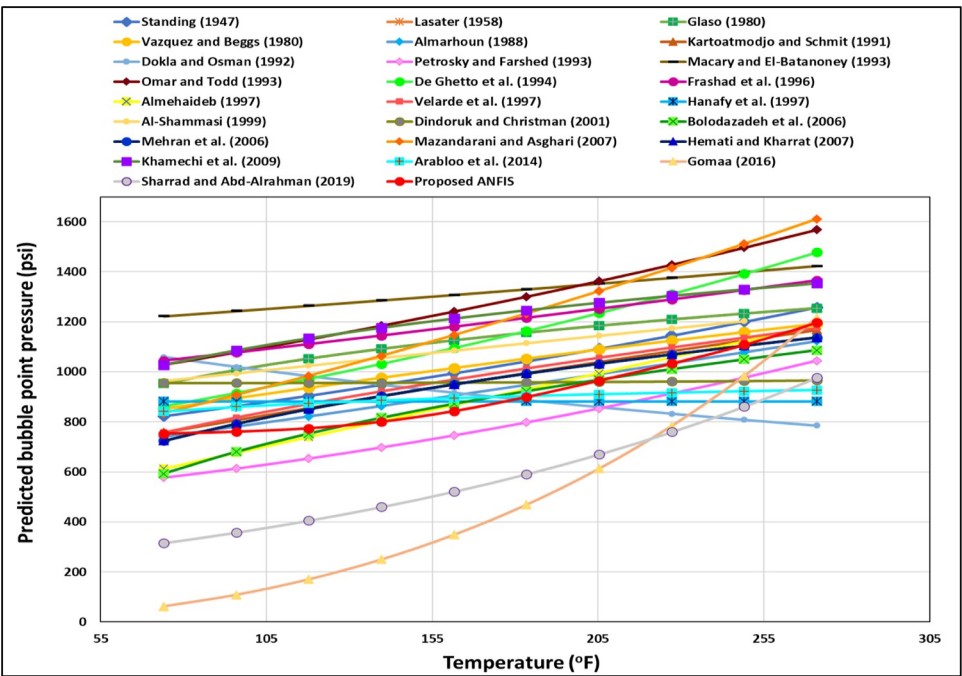

**Fig 11.** $T_f$ TrA of the ANFIS and existing models.

## 3.2 Comparison of the ANFIS model against other models

**3.2.1 Cross-plot.** Fig 13 shows the cross-plot for the training datasets of the ANFIS model. Most training data are closer to the 45˚ line to indicate that the ANFIS is a higher accurate model for the training datasets. The ($R^2$) for the training datasets of the ANFIS model is

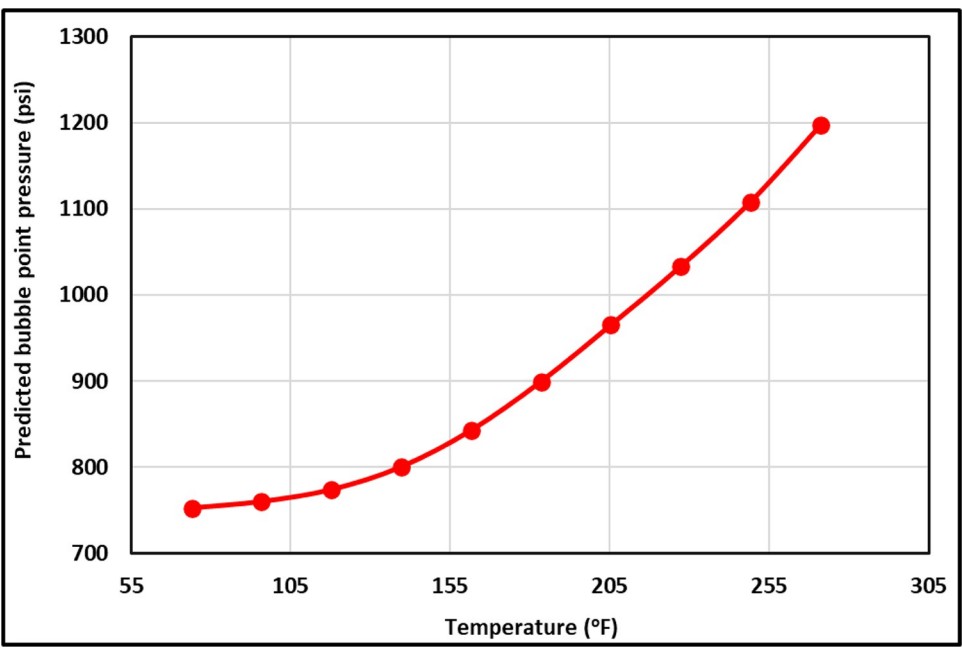

**Fig 12.** $T_f$ TrA of the ANFIS model.

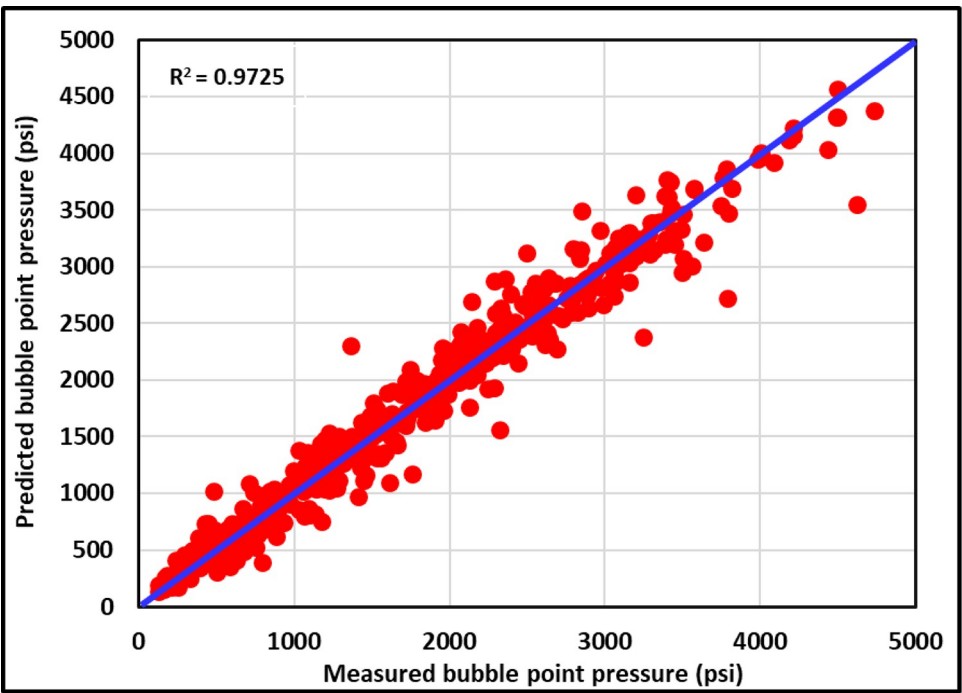

**Fig 13. Cross-plot of training ANFIS model.**

0.9725. Fig 14 presents the cross plot for the testing datasets of the ANFIS model, and most of the testing data are also closer to the 45˚ line to show that the ANFIS model can accurately predict the $P_b$ for the testing datasets with the ($R^2$) of 0.9878. Fig 15 displays the cross-plot for the

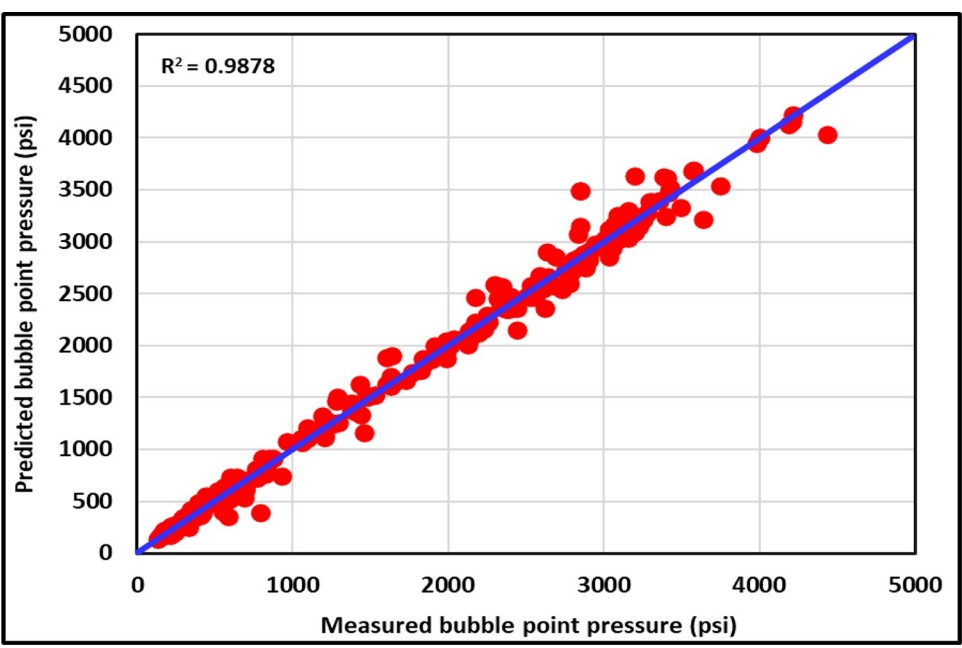

**Fig 14. Cross-plot of testing ANFIS model.**

ANFIS and all current models studied in this paper. As shown in Fig 15, the ANFIS model is the highest accurate model with ($R^2$) of 0.9878 compared to all studied models.

**3.2.2 Statistical error analysis.** Some statistical analysis has been used along with trend analysis and cross-plotting analysis to validate and describe the efficiencies of the proposed ANFIS model. In addition, the ANFIS was compared against the 22 studied models that follow the correct physical behavior. The statistical error analysis applying in this study are (R), RMSE, SD, APRE, AAPRE, maximum and minimum absolute percent relative error ($E_{max.}$) and ($E_{min.}$). The statistical criterion explanations are presented in the appendix (S1 Appendix). The AAPRE and R were used in this research as the leading indicators to compare the ANFIS model's accuracy with the current models.

The ANFIS and existing models were compared by plotting the AAPRE and R (Fig 16). As display in Fig 16, the ANFIS model is the first rank model and has the lowest AAPRE of 6.378% and APRE of -0.99%, and the highest (R) of 0.994. The second rank model is Velarde et al.'s [4] model with the AAPRE of 9%, the APRE of -1.58%, and R of 0.9724. The third rank model is Mehran et al.'s [14] correlation with the AAPRE of 9.75%, the APRE of -3.91%, and R of 0.9699. The last rank model is Petrosky and Farshad's [8] model with the AAPRE of 76.59%, the APRE of 57.39%, and R of 0.9703.

The ANFIS and all existing models are compared using statistical error analyses AAPRE, APRE, RMSE, SD, $E_{min.}$, and $E_{max.}$, Table 4. The ANFIS model and all studied models are ranked based on the leading indicators AAPRE and R. The ANFIS model is the first rank model and has the lowest AAPRE of 6.38%, APRE of -0.99, RMSE of 9.73, SD of 0.074, $E_{min.}$of 0.021%, and $E_{max.}$ of 50.19% and the highest R of 0.9939. The results indicate that the ANFIS model outperformed all existing models (22 models). The second rank model is Velarde et al.'s [4] correlation that has the AAPRE of 9%, APRE of -1.58, RMSE of 13.04, SD of 0.094, $E_{min.}$ of

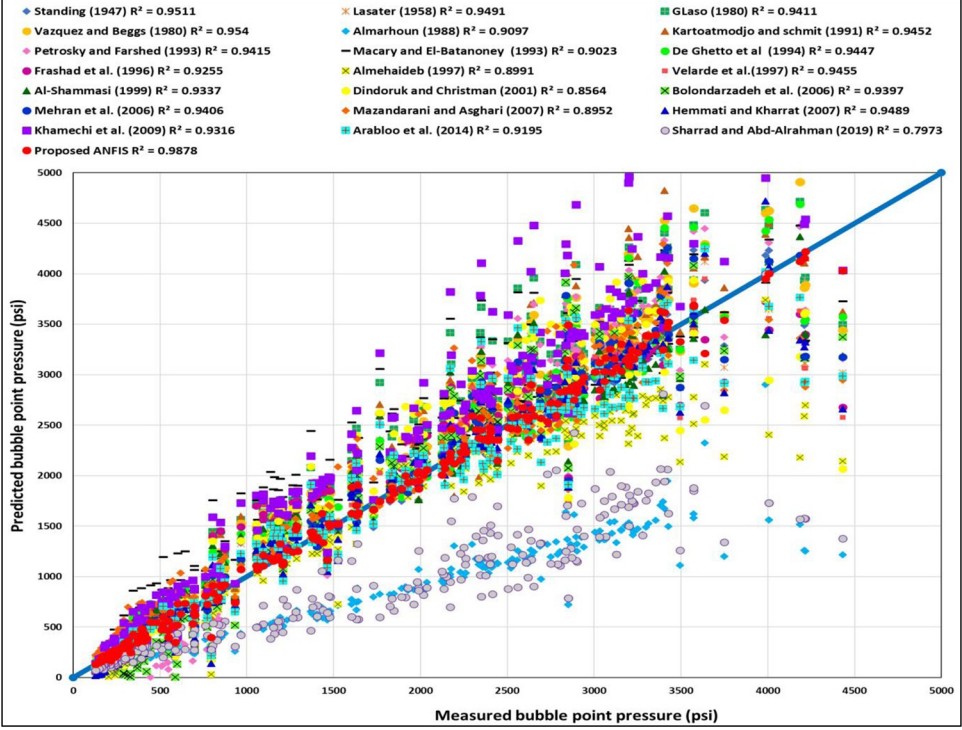

**Fig 15. Cross-plot of the ANFIS and existing models.**

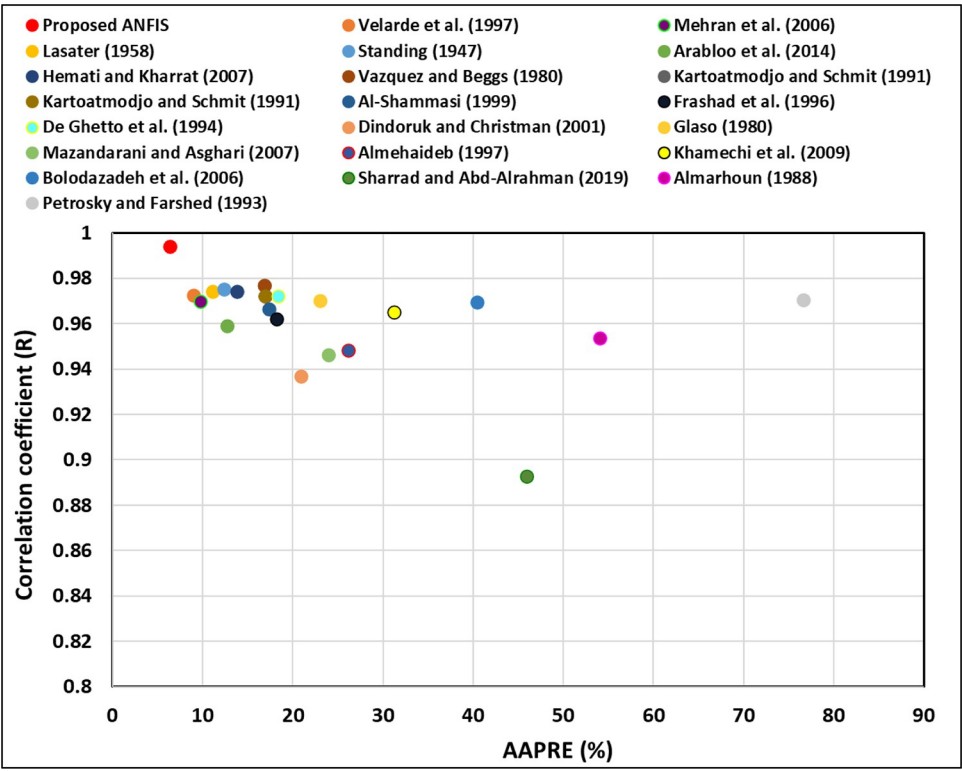

**Fig 16. Comparing the ANFIS and existing models using (R) and AARE (%).**

0.039, $E_{max.}$ of 62.47, and R of 0.9724. The third rank model is Mehran et al.'s [14] correlation and has the AAPRE of 9.75%, APRE of -3.91%, RMSE of 13.60, SD of 0.095, $E_{min.}$ of 0.035%, $E_{max.}$ of 63.86%, and R of 0.9699. The last rank model is Petrosky and Farshed's [8] correlation that has the AAPRE of 76.59%, APRE of 57.39%, RMSE of 159.87, SD of 1.406, $E_{min.}$ of 0.295%, $E_{max.}$ of 784.59%, and R of 0.9703. Comparing the ANFIS and existing models conducts an important means of evaluating all the models' performance.

## 4. Conclusions

With 760 global datasets used, the ANFIS model was developed with the trend analysis to robustly and accurately predict the $P_b$. In addition, the ANFIS mode's accuracy was compared with 21 existing models utilizing statistical error analysis. In this research, we can conclude the following:

- The trend analysis results of the ANFIS model indicate that the ANFIS model can describe the correct relationships between the independent parameters (Rs, $\gamma_g$, API, $T_f$) and dependent parameter $P_b$ to show the proper physical behavior.

- Some previous correlations fail to represent the proper relationships between the independent parameters and the $P_b$ to indicate incorrect physical behavior.

- The proposed ANFIS model outperformed all 21 existing models and has the lowest AAPRE of 6.38%, APRE of -0.99, RMSE of 9.73, SD of 0.074, $E_{min.}$ of 0.021%, and $E_{max.}$ of 50.19% and the highest R of 0.9939 compared to 21 studied correlations that follow the correct physical behavior. The ANFIS model shows better results than other models because of its

**Table 4. Statistical error analysis of the ANFIS and existing models.**

| Rank | Model | APRE (%) | AAPRE (%) | $E_{max.}$ (%) | $E_{min.}$ (%) | RMSE (psi) | SD (psi) | R |
|---|---|---|---|---|---|---|---|---|
| 1 | **Proposed ANFIS** | -0.99 | 6.38 | 50.19 | 0.021 | 9.73 | 0.074 | 0.9939 |
| 2 | **Velarde et al. (1997) [4]** | -1.58 | 9.00 | 62.47 | 0.039 | 13.04 | 0.095 | 0.9724 |
| 3 | **Mehran et al. (2006) [14]** | -3.91 | 9.75 | 63.86 | 0.035 | 13.60 | 0.095 | 0.9699 |
| 4 | **Lasater (1958) [6]** | -1.83 | 11.07 | 66.08 | 0.016 | 15.31 | 0.106 | 0.9742 |
| 5 | **Standing (1947) [5]** | -3.95 | 12.35 | 69.28 | 0.032 | 16.26 | 0.106 | 0.9753 |
| 6 | **Arabloo et al. (2014) [28]** | 1.51 | 12.66 | 72.98 | 0.000 | 17.12 | 0.116 | 0.9589 |
| 7 | **Hemati and Kharrat (2007) [16]** | 6.35 | 13.76 | 85.01 | 0.026 | 22.13 | 0.174 | 0.9741 |
| 8 | **Vazquez and Beggs (1980) [25]** | -13.07 | 16.88 | 74.79 | 0.493 | 21.65 | 0.136 | 0.9767 |
| 9 | **Kartoatmodjo and Schmit (1991) [26]** | -9.33 | 16.94 | 78.37 | 0.085 | 22.74 | 0.152 | 0.9722 |
| 10 | **Al-Shammasi (1999) [27]** | -11.20 | 17.33 | 62.95 | 0.205 | 22.60 | 0.145 | 0.9663 |
| 11 | **Frashad et al. (1996) [23]** | -8.03 | 18.23 | 74.23 | 0.042 | 24.30 | 0.161 | 0.9621 |
| 12 | **De Ghetto et al. (1994) [9]** | -14.18 | 18.37 | 73.97 | 0.007 | 24.83 | 0.167 | 0.9720 |
| 13 | **Dindoruk and Christman (2001) [10]** | -3.72 | 20.89 | 77.83 | 0.432 | 25.81 | 0.152 | 0.9369 |
| 14 | **Glaso (1980) [7]** | -14.33 | 23.02 | 79.52 | 0.281 | 27.70 | 0.154 | 0.9701 |
| 15 | **Mazandarani and Asghari (2007) [17]** | -19.19 | 23.91 | 120.93 | 0.127 | 34.19 | 0.245 | 0.9462 |
| 16 | **Almehaideb (1997) [13]** | 22.89 | 26.15 | 234.92 | 0.037 | 44.18 | 0.357 | 0.9482 |
| 17 | **Macary and El-Batanoney (1993) [20]** | -25.03 | 31.20 | 149.75 | 0.111 | 42.62 | 0.291 | 0.9499 |
| 18 | **Khamechchi et al. (2009) [18]** | -29.55 | 31.24 | 97.52 | 0.059 | 37.27 | 0.204 | 0.9652 |
| 19 | **Bolodarzadeh et al. (2006) [15]** | 28.31 | 40.42 | 434.20 | 0.175 | 84.69 | 0.746 | 0.9694 |
| 20 | **Sharrad and Abd-Alrahman (2019) [22]** | 45.92 | 45.93 | 72.46 | 0.346 | 47.96 | 0.139 | 0.8929 |
| 21 | **Al-marhoun (1988) [11]** | 54.06 | 54.06 | 79.22 | 27.176 | 54.40 | 0.148 | 0.9538 |
| 22 | **Petrosky and Farshed (1993) [8]** | 57.39 | 76.59 | 784.59 | 0.295 | 159.87 | 1.406 | 0.9703 |

combination of the FL and ANN performances and better learning ability. The ANFIS can perform a highly non-linear mapping.

• The data randomization was conducted to prevent the model from overfitting or underfitting to obtain the robust and accurate ANFIS model to predict the $P_b$.

## Supporting information

**S1 Appendix.**
(PDF)

## Acknowledgments

Special thanks to the Centre of Research in Enhanced Oil Recovery (COREOR), Petroleum Engineering department, Universiti Teknologi PETRONAS for supporting this work.

## Author Contributions

**Conceptualization:** Fahd Saeed Alakbari, Mysara Eissa Mohyaldinn.

**Data curation:** Fahd Saeed Alakbari.

**Formal analysis:** Fahd Saeed Alakbari, Ibnelwaleed A. Hussein.

**Funding acquisition:** Mysara Eissa Mohyaldinn.

**Investigation:** Mohammed Abdalla Ayoub.

**Methodology:** Fahd Saeed Alakbari, Mohammed Abdalla Ayoub.

**Project administration:** Mysara Eissa Mohyaldinn.

**Software:** Fahd Saeed Alakbari, Mohammed Abdalla Ayoub.

**Supervision:** Mysara Eissa Mohyaldinn, Ali Samer Muhsan, Ibnelwaleed A. Hussein.

**Visualization:** Mysara Eissa Mohyaldinn, Ibnelwaleed A. Hussein.

**Writing – original draft:** Fahd Saeed Alakbari.

**Writing – review & editing:** Mysara Eissa Mohyaldinn, Mohammed Abdalla Ayoub, Ali Samer Muhsan, Ibnelwaleed A. Hussein.

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
