## [Decision Letter · Decision Letter 0]

1 May 2022

PONE-D-22-03484A reservoir bubble point pressure prediction model using Adaptive Neuro-Fuzzy Inference System (ANFIS) technique with trend analysisPLOS ONE

Dear Dr. Alakbari,

Thank you for submitting your manuscript to PLOS ONE. After careful consideration, we feel that it has merit but does not fully meet PLOS ONE’s publication criteria as it currently stands. Therefore, we invite you to submit a revised version of the manuscript that addresses the points raised during the review process.

We look forward to receiving your revised manuscript.

Kind regards,

Ardashir Mohammadzadeh, Phd

Academic Editor

PLOS ONE

Journal Requirements:

4. Thank you for stating the following in the Funding Section of your manuscript:

“The authors would like to give their heartfelt thanks to the Yayasan Universiti Teknologi PETRONAS (YUTP FRG Grant No: 015LC0-226) at Universiti Teknologi PETRONAS for supporting this study.”

“The authors would like to give their heartfelt thanks to the Yayasan Universiti Teknologi PETRONAS (YUTP FRG Grant No: 015LC0-226) at Universiti Teknologi PETRONAS for supporting this study.”

“The authors would like to give their heartfelt thanks to the Yayasan Universiti Teknologi PETRONAS (YUTP FRG Grant No: 015LC0-226) at Universiti Teknologi PETRONAS for supporting this study.”

6. We note that Figure 1 in your submission contain [map/satellite] images which may be copyrighted. All PLOS content is published under the Creative Commons Attribution License (CC BY 4.0), which means that the manuscript, images, and Supporting Information files will be freely available online, and any third party is permitted to access, download, copy, distribute, and use these materials in any way, even commercially, with proper attribution. For these reasons, we cannot publish previously copyrighted maps or satellite images created using proprietary data, such as Google software (Google Maps, Street View, and Earth). For more information, see our copyright guidelines: http://journals.plos.org/plosone/s/licenses-and-copyright.

7. Please include captions for your Supporting Information files at the end of your manuscript, and update any in-text citations to match accordingly. Please see our Supporting Information guidelines for more information: http://journals.plos.org/plosone/s/supporting-information

Reviewers' comments:

Reviewer's Responses to Questions

**Comments to the Author**

1. Is the manuscript technically sound, and do the data support the conclusions?

Reviewer #1: Yes

Reviewer #2: Partly

2. Has the statistical analysis been performed appropriately and rigorously? 

Reviewer #1: Yes

Reviewer #2: Yes

3. Have the authors made all data underlying the findings in their manuscript fully available?

Reviewer #1: Yes

Reviewer #2: Yes

4. Is the manuscript presented in an intelligible fashion and written in standard English?

Reviewer #1: No

Reviewer #2: Yes

5. Review Comments to the Author

Reviewer #1: The abstract needs a little bit more numerical result. Please add them.

Please explain why did you chose ANFIS among other models? Why did not you use ANN, SVM, etc?

Page 3, line 59: please use a uniform mode when reporting the numbers. Please correct “one hundred and 158 datasets”.

Considering the fact that your research only uses ANFIS, I suggest you to explain more about ANFIS including its steps and equations. Use following papers for that.

Azad, A., Farzin, S., Sanikhani, H., Karami, H., Kisi, O., & Singh, V. P. (2021). Approaches for optimizing the performance of adaptive neuro-fuzzy inference system and least-squares support vector machine in precipitation modeling. Journal of Hydrologic Engineering, 26(4), 04021010.

Azad, A., Manoochehri, M., Kashi, H., Farzin, S., Karami, H., Nourani, V., & Shiri, J. (2019). Comparative evaluation of intelligent algorithms to improve adaptive neuro-fuzzy inference system performance in precipitation modelling. Journal of hydrology, 571, 214-224.

Please change Table 3 to a casual table in terms of colour.

How did you select the setting parameters of ANFIS. For instance, what was the reason training epoch is 10?

I think figure 4 has been copied from another article. Please cite them and ask for their permission.

The novelty of this work is not clear enough yet. I mean, it should be cleared that how much ANFIS improved the modeling accuracy in comparison with other simpler models? Why we need to use ANFIS?

Results and discussion is satisfying.

We do not usually cite other papers in the Conclusion section. Please remove them and modify the conclusion accordingly.

Reviewer #2: Please give explanation why the developed ANFIS gave low or high values of reservoir bubble point pressure

when compared against the 22 studied models that follow the correct physical behavior.

6. PLOS authors have the option to publish the peer review history of their article (what does this mean?). If published, this will include your full peer review and any attached files.

Reviewer #1: No

Reviewer #2: No

---

## [Author Response · Author response to Decision Letter 0]

6 Jul 2022

Reviewer #1: 

The abstract needs a little bit more numerical result. Please add them.

Done, the numerical results were added to the abstract accordingly.

Please explain why did you choose ANFIS among other models? Why did not you use ANN, SVM, etc?

The ANFIS technique has the advantage of showing better results than other methods, which is why it is chosen in this study. The ANFIS offers a better learning ability. It can perform a highly nonlinear mapping. It has fewer adjustable parameters than those needed in other machine learning. Its structure can allow for parallel computation. Its networks show a well-structured knowledge representation and can also allow better integration with other control design methods [1]. ANFIS can combine ANN and Fl in a single tool to make the technique superb in reaching a quicker decision about the mapped relationship between the feature and target parameters [2]. The ANFIS benefits from decreased training time because of its smaller dimensions and because the network is initialized with parameters about the problem domain [3]. 

Page 3, line 59: please use a uniform mode when reporting the numbers. Please correct “one hundred and 158 datasets”.

Done, the numbers were corrected accordingly. 

Considering the fact that your research only uses ANFIS, I suggest you to explain more about ANFIS, including its steps and equations. Use the following papers for that.

Azad, A., Farzin, S., Sanikhani, H., Karami, H., Kisi, O., & Singh, V. P. (2021). Approaches for optimizing the performance of adaptive neuro-fuzzy inference system and least-squares support vector machine in precipitation modeling. Journal of Hydrologic Engineering, 26(4), 04021010.

Azad, A., Manoochehri, M., Kashi, H., Farzin, S., Karami, H., Nourani, V., & Shiri, J. (2019). Comparative evaluation of intelligent algorithms to improve adaptive neuro-fuzzy inference system performance in precipitation modelling. Journal of Hydrology, 571, 214-224.

Done, more explanation about ANFIS, including its steps and equations, is added accordingly using the recommended papers. Please, you can see highlighted yellow color in subsection (2.2 Proposed ANFIS model strategy).

Please change Table 3 to a casual table in terms of colour.

Done, the Table was changed to a casual table in terms of color accordingly.

How did you select the setting parameters of ANFIS? For instance, what was the reason the training epoch is 10?

The optimal hyper-parameters of ANFIS were selected by using the manual method. In the manual way, each parameter changed in its different types or values. Then, the model accuracy and the correct trend analysis were checked. Finally, the optimal hyper-parameters were selected with the appropriate trend analysis for the highest accuracy, as shown in Table 3. The statement is added in subsection 2.2 Proposed ANFIS model strategy accordingly.

I think figure 4 has been copied from another article. Please cite them and ask for their permission.

Figure 4 was generated from MATLAB R2019b; it is mentioned accordingly. 

The novelty of this work is not clear enough yet. It should be clear how much ANFIS improved the modelling accuracy compared to cleared and how much ANFIS improved the modeling accuracy compared to the other simpler models? 

The novelty of this study is to apply the ANFIS model with the trend analysis. The trend analysis study shows that the model follows the correct relationships between the inputs and output to prove the proper physical behavior. In addition, the ANFIS model is higher accuracy than all studied models. Please, see the added statement in the abstract accordingly. The second rank model that is the best in the previous models has R, AAPRE, APRE, SD, and RMSE of 0.9724, 9%, -1.58%, 0.095 psi, and 13.04 psi, respectively. However, the proposed ANFIS model has the highest R of 0.994 and the lowest AAPRE, APRE, SD, and RMSE of 6.38%, -0.99%, 0.074 psi, and 9.73 psi, respectively, as the first rank model. These values are added in the abstract accordingly. 

Why we need to use ANFIS?

It is the first time to use the ANFIS model with the trend analysis to determine the Pb because the ANFIS shows better results as discussed in the advantages of the ANFIS in the subsection (2.2 Proposed ANFIS model strategy). In addition, the trend analysis is used to validate the ANFIS model to follow the correct relationships between the inputs and output to prove the proper physical behavior or to show the effects of the inputs on the output. 

Results and discussion are satisfying.

Thank you a lot. 

We do not usually cite other papers in the Conclusion section. Please remove them and modify the conclusion accordingly.

Done, the conclusion was modified accordingly.

References

[1] R. Isanta Navarro, “Study of a neural network-based system for stability augmentation of an airplane,” 2013.

[2] P. Tahmasebi and A. Hezarkhani, “A hybrid neural networks-fuzzy logic-genetic algorithm for grade estimation,” Comput. Geosci., vol. 42, pp. 18–27, 2012.

[3] L. P. Maguire, B. Roche, T. M. McGinnity, and L. J. McDaid, “Predicting a chaotic time series using a fuzzy neural network,” Inf. Sci. (NY)., vol. 112, no. 1–4, pp. 125–136, 1998.

Reviewer #2: 

Please give explanation why the developed ANFIS gave low or high values of reservoir bubble point pressure when compared against the 22 studied models that follow the correct physical behavior.

The ANFIS model shows better results than other models because of combining the FL and ANN performances and better learning ability. The ANFIS can perform a highly non-linear mapping. These statements are added to the conclusion accordingly. The other advantages of the ANFIS are added in the subsection (2.2 Proposed ANFIS model strategy).

---

## [Decision Letter · Decision Letter 1]

27 Jul 2022

A reservoir bubble point pressure prediction model using the Adaptive Neuro-Fuzzy Inference System (ANFIS) technique with trend analysis

PONE-D-22-03484R1

Dear Dr. Alakbari,

We’re pleased to inform you that your manuscript has been judged scientifically suitable for publication and will be formally accepted for publication once it meets all outstanding technical requirements.

Kind regards,

Ardashir Mohammadzadeh, Phd

Academic Editor

PLOS ONE

Additional Editor Comments (optional):

Reviewers' comments:

Reviewer's Responses to Questions

**Comments to the Author**

1. If the authors have adequately addressed your comments raised in a previous round of review and you feel that this manuscript is now acceptable for publication, you may indicate that here to bypass the “Comments to the Author” section, enter your conflict of interest statement in the “Confidential to Editor” section, and submit your "Accept" recommendation.

Reviewer #1: All comments have been addressed

2. Is the manuscript technically sound, and do the data support the conclusions?

Reviewer #1: Yes

3. Has the statistical analysis been performed appropriately and rigorously? 

Reviewer #1: Yes

4. Have the authors made all data underlying the findings in their manuscript fully available?

Reviewer #1: Yes

5. Is the manuscript presented in an intelligible fashion and written in standard English?

Reviewer #1: Yes

6. Review Comments to the Author

Reviewer #1: I think the authors addressed my concerns carefully and the manuscript is ready to be published now.

7. PLOS authors have the option to publish the peer review history of their article (what does this mean?). If published, this will include your full peer review and any attached files.

Reviewer #1: No

---

## [Editor Report · Acceptance letter]

1 Aug 2022

PONE-D-22-03484R1 

A reservoir bubble point pressure prediction model using the Adaptive Neuro-Fuzzy Inference System (ANFIS) technique with trend analysis 

Dear Dr. Alakbari:

I'm pleased to inform you that your manuscript has been deemed suitable for publication in PLOS ONE. Congratulations! Your manuscript is now with our production department. 

Kind regards, 

on behalf of

Dr. Ardashir Mohammadzadeh 

Academic Editor

PLOS ONE